 **eLIFE**

# Reconstitution of bacterial autotransporter assembly using purified components

Giselle Roman-Hernandez, Janine H Peterson, Harris D Bernstein*

Genetics and Biochemistry Branch, National Institute of Diabetes and Digestive and Kidney Diseases, National Institutes of Health, Bethesda, United States

**Abstract** Autotransporters are a superfamily of bacterial virulence factors consisting of an N-terminal extracellular ('passenger') domain and a C-terminal β barrel ('β') domain that resides in the outer membrane (OM). The mechanism by which the passenger domain is secreted is poorly understood. Here we show that a conserved OM protein insertase (the Bam complex) and a molecular chaperone (SurA) are both necessary and sufficient to promote the complete assembly of the *Escherichia coli* O157:H7 autotransporter EspP in vitro. Our results indicate that the membrane integration of the β domain is the rate-limiting step in autotransporter assembly and that passenger domain translocation does not require the input of external energy. Furthermore, experiments using nanodiscs strongly suggest that autotransporter assembly is catalyzed by a single copy of the Bam complex. Finally, we describe a method to purify a highly active form of the Bam complex that should facilitate the elucidation of its function.

## Introduction

The autotransporter (type Va) pathway is the most widespread virulence factor secretion pathway in Gram-negative bacteria (*Leyton et al., 2012*). Autotransporters are single polypeptides that consist of two domains, an N-terminal domain ('passenger domain') that is exposed on the cell surface and that often mediates a virulence function, and a C-terminal domain ('β domain') that resides in the outer membrane (OM). Following their translocation across the OM, many passenger domains are released from the cell surface by a proteolytic cleavage. Passenger domains vary widely in sequence and size (~20–400 kD), but almost always form an unusual repetitive structure known as a β helix (*Junker et al., 2006*). β domains are typically ~30 kD in size and, like the vast majority of bacterial integral OM proteins, fold into a β barrel structure. While the sequences of β domains are also very heterogeneous, the structures of all of the β domains that have been solved to date are nearly superimposable (*Oomen et al., 2004*; *Barnard et al., 2007*; *van den Berg, 2010*; *Zhai et al., 2011*). After autotransporters are translocated across the inner membrane (IM) through the Sec complex they interact with molecular chaperones that presumably maintain them in an assembly-competent state (*Ieva and Bernstein, 2009*; *Ruiz-Perez et al., 2009*; *Ieva et al., 2011*). Subsequently the β domain is targeted to the Bam complex, an essential heterooligomer consisting of an integral OM protein (BamA) and four lipoproteins (BamBCDE) that catalyzes the membrane insertion of β barrel proteins (*Voulhoux et al., 2003*; *Wu et al., 2005*; *Sauri et al., 2009*; *Hagan et al., 2010*; *Ieva et al., 2011*). X-ray crystallographic analysis suggests that the BamA β barrel domain is unstable and may open laterally to allow the escape of client proteins into the lipid bilayer (*Noinaj et al., 2013*, *2014*).

Although it has been shown that the passenger domain is translocated across the OM in a C-to-N-terminal fashion (*Ieva and Bernstein, 2009*; *Junker et al., 2009*), the mechanism of translocation is poorly understood. Based on the observation that deletion of the β domain abolishes passenger domain translocation, it was originally proposed that the β domain serves as the transport channel for the passenger domain (whence the name 'autotransporter') (*Pohlner et al., 1987*). At first glance, this

*For correspondence: harris_bernstein@nih.gov

**Competing interests:** The authors declare that no competing interests exist.

**eLife digest** Disease-causing bacteria release molecules called virulence factors to help them infect their host. These virulence factors need to pass through the membrane that surrounds the cell. Indeed, some bacteria, such as *Escherichia coli*, have two membranes, so some virulence factors need to pass through an extra membrane.

One group of virulence factors found in *E. coli* are called autotransporters. These proteins have two sections: the passenger domain, which is the main part of the virulence factor, and the β domain, which anchors the autotransporter in the outer membrane. Once the passenger domain is outside the cell, the link to the β domain can be broken to release the virulence factor. However, we do not know how the passenger domain passes through the outer membrane.

By studying an *E. coli* autotransporter called EspP, Roman-Hernandez et al. have now identified the other proteins that are required for the β domain to insert into an artificial membrane, and allow the passenger domain to pass through the membrane. These other proteins are a group of proteins called the Bam complex and a chaperone protein called SurA. The experiments also show that an external source of energy is not needed to drive this process, and they suggest that the passenger domain moves through a hole in the outer membrane formed by the β domain and/or the Bam complex. Roman-Hernandez et al. also developed a new way to purify the Bam complex that should help all researchers working on this set of proteins.

hypothesis seems to be supported by crystallographic studies showing that upon completion of translocation the two domains are connected by an α-helical linker that is embedded inside the β domain pore (*Oomen et al., 2004*; *Barnard et al., 2007*; *van den Berg, 2010*). The same studies, however, revealed that the β domain pore is only ~10 Å in diameter. Because the directionality of translocation presumably requires the formation of a C-terminal hairpin followed by the sliding of medial and N-terminal segments past a static strand, both strands of the hairpin would have to be in a fully extended conformation to fit inside the pore. On the contrary, available evidence indicates that the polypeptide that resides inside the β domain forms an α helix prior to the completion of translocation (*Ieva et al., 2008*; *Peterson et al., 2010*). Furthermore, small folded polypeptides have been shown to be secreted efficiently by the autotransporter pathway and a subset of naturally occurring passenger domains undergo disulfide bonding in the periplasm (*Skillman et al., 2005*; *Jong et al., 2007*). Finally, BamA has been shown to interact with the passenger domain during the translocation reaction (*Ieva and Bernstein, 2009*). While the β domain does appear to play a role in translocation (*Saurí et al., 2011*; *Pavlova et al., 2013*), these and other results strongly suggest that autotransporter assembly is more complex than originally envisioned. In one scenario, the Bam complex might facilitate the membrane insertion of the β domain and the translocation of the passenger domain in a concerted reaction, but it is also possible that other factors are also required to transport the passenger domain across the OM.

The energy source for passenger domain translocation has also remained poorly understood. Although most protein translocation reactions require external energy in the form of ATP or a membrane potential, the periplasm is devoid of ATP and there is no membrane potential across the bacterial OM. It is conceivable that an IM protein that derives energy from the hydrolysis of cytoplasmic ATP or from the electrochemical gradient across the IM interacts with the passenger domain and drives translocation, but such a protein has never been identified. To explain the energetics of secretion, it has been proposed that the sequential folding of segments of the β helix in the extracellular space drives translocation and prevents retrograde movement into the periplasm (*Klauser et al., 1992*; *Junker et al., 2006*). Consistent with this vectorial folding model, mutations that impair the folding of C-terminal segments of two different passenger domains significantly reduce the efficiency of secretion (*Peterson et al., 2010*; *Renn et al., 2012*). Mutations that perturb the folding of medial and N-terminal segments of the passenger domain, however, produce only a modest translocation defect (*Kang'ethe and Bernstein, 2013b*). Furthermore, the intrinsically disordered receptor domain (RD) of the *Bordetella pertussis* RTX toxin has been shown to be secreted efficiently by the autotransporter pathway (*Kang'ethe and Bernstein, 2013a*). Interestingly, the presence of a large number of acidic

residues is critical for the secretion of the RD. This observation, together with the finding that naturally occurring passenger domains are predominantly acidic, suggests that charge interactions may play a significant role in driving the translocation reaction.

To gain further insight into both the mechanism and energetics of passenger domain translocation, we sought to reconstitute autotransporter assembly in vitro using purified components. In our experiments, we used the *Escherichia coli* O157:H7 autotransporter EspP, as a model protein. Following the completion of translocation, the passenger domain of EspP and other members of the SPATE (serine protease autotransporters of Enterobacteriaceae) family is released in an unusual intrabarrel cleavage reaction that requires accurate folding of the β domain and precise positioning of the passenger domain-β domain junction (*Dautin et al., 2007*; *Barnard et al., 2012*). Although it was previously shown that the insertion of small *E. coli* β barrel proteins such as OmpT and OmpA into proteoliposomes can be catalyzed by the purified Bam complex and the periplasmic chaperone SurA (*Hagan et al., 2010*; *Hagan and Kahne, 2011*; *Hagan et al., 2013*), we did not observe assembly of EspP using the same methodology. Using an alternative expression and purification strategy, however, we obtained an apparently more active form of the Bam complex that, together with SurA, was both necessary and sufficient to promote the membrane integration of the EspP β domain and the translocation and cleavage of the EspP passenger domain. Remarkably, passenger domain translocation did not require the input of any additional energy. In addition to defining the minimal set of factors required for autotransporter assembly, our work provides a valuable resource for future studies on the function of the Bam complex and its role in the biogenesis of the broader class of OM proteins.

## Results

### The Bam complex and SurA are necessary and sufficient to promote the assembly of an EspP derivative that contains a minimal passenger domain fragment

It has long been known that large N-terminal segments of autotransporter passenger domains can be deleted without affecting the integration of the β domain into the OM or the secretion of the remaining passenger domain fragment (*Dautin and Bernstein, 2007*). The native EspP passenger domain is 968 residues in length (*Figure 1A*), but a derivative that contains only 26 residues of the ~28 residue C-terminal segment that reside inside the β domain in an α-helical conformation (EspPΔ5) is assembled as efficiently as the wild-type protein in vivo (*Pavlova et al., 2013*). The modification or deletion of residues in the α-helical segment, however, can profoundly perturb the folding and integration of the β domain into the OM and/or the proteolytic release of the passenger domain, which requires precise alignment of the cleavage junction with key catalytic residues (*Dautin et al., 2007*; *Ieva et al., 2008*; *Barnard et al., 2012*). Because EspPΔ5 and other derivatives that expose no more than a short segment on the cell surface are structurally similar to generic β barrel proteins such as OmpT, it might be expected that they would have similar assembly requirements. Indeed any autotransporter-specific assembly factors might only be required once the passenger domain reaches a threshold size. Based on this reasoning, we first analyzed the assembly of an EspP derivative designated EspP(46+β) that consists of EspPΔ5 plus 20 N-terminal residues derived from the cloning vector (*Figure 1B*). We expected that the accurate assembly of EspP(46+β) would result in the proteolytic release of the 46 residue passenger domain and the accumulation of a folded ~30 kD β domain that, as previously shown, is resistant to SDS denaturation unless heated (*Skillman et al., 2005*).

Initially we tested whether EspP(46+β) would assemble in vitro using a previously described OM protein assembly assay (*Hagan et al., 2010*). In this assay, *E. coli* BamAB and BamCDE are first expressed and isolated independently. The two subcomplexes are then mixed together to form a holocomplex and reconstituted into proteoliposomes. In the presence of the proteoliposomes and a molar excess of SurA, denatured OmpT folds into a stable structure that is enzymatically active and resistant to SDS denaturation. We attempted to express and purify the Bam complex exactly as described, but for reasons that are unclear we did not observe efficient formation of the Bam holocomplex unless we purified BamAB through an additional step. Nevertheless, we ultimately obtained a single peak on a gel filtration column that was highly enriched in the reconstituted complex, which we designated Bam(AB)(CDE) (*Figure 2—figure supplement 1A,B*). After the peak fractions were pooled,

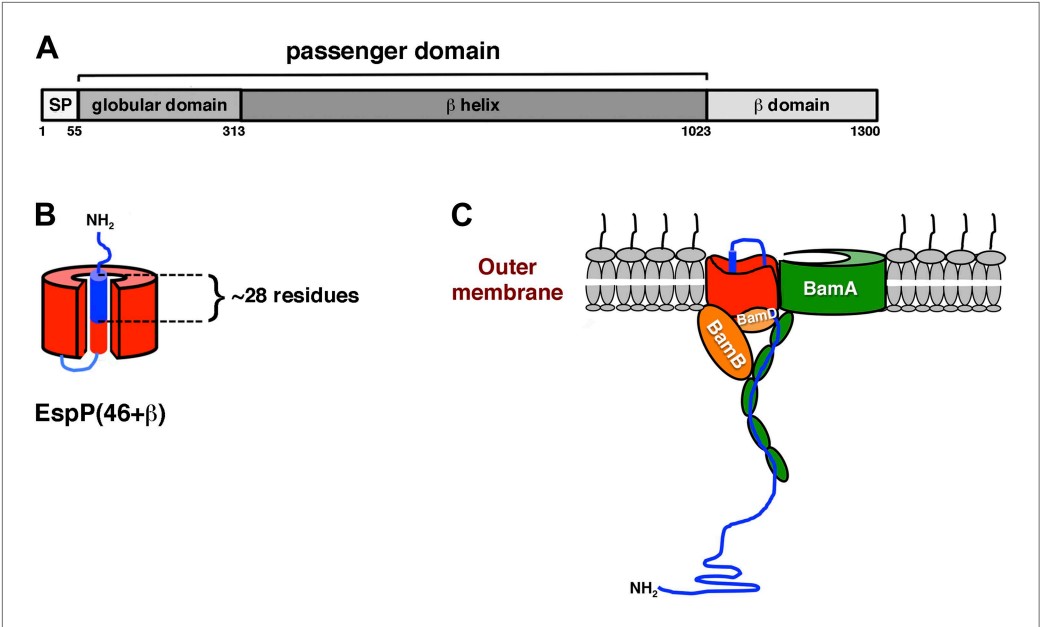

**Figure 1**. Domain structure of EspP and model for passenger domain translocation. (**A**) EspP consists of a signal peptide (SP; residues 1–55), an extracellular ('passenger') domain (residues 56–1023) and a β barrel ('β') domain (residues 1024–1300) (**Brunder et al., 1997**). While most of the passenger domain forms a long β helix, the N-terminus (residues 56–313) forms a discrete globular domain (**Khan et al., 2011**). (**B**) Illustration of EspP(46+β). Prior to the release of the passenger domain in an intrabarrel cleavage reaction, ~28 residues of the 46 residue passenger domain are embedded inside the β domain pore. (**C**) Available evidence indicates that the EspP passenger domain is secreted in a hairpin conformation while distinct regions of the β domain interact with BamA, BamB and BamD (**Ieva et al., 2011**). Components of the transport channel likely include the open β domain and/ or the BamA β barrel, which has been proposed to open laterally (**Noinaj et al., 2013**, **2014**). In any case, the finding that proteolytic maturation and the release of the β domain from the Bam complex both require the completion of translocation (**Peterson et al., 2010**; **Ieva et al., 2011**; **Pavlova et al., 2013**) strongly suggests that the active site cannot form during the passage of the passenger domain across the OM. BamC and BamE hve been omitted from the model for the sake of clarity.

most of the Bam holocomplex could be reconstituted into proteoliposomes (**Figure 2—figure supplement 1C**). Subsequently we purified SurA to homogeneity as described (**Hagan et al., 2010**) (**Figure 2—figure supplement 2**). To assess the folding of OmpT, we used an activity assay in which the cleavage of a fluorogenic peptide leads to an increase in fluorescence intensity over time. Consistent with previous results, we observed a fluorescent signal when OmpT was incubated in the presence of Bam(AB)(CDE) and SurA (**Figure 2E**, purple curve). EspP(46+β) purified from inclusion bodies (**Figure 3—figure supplement 1**) was mixed with the same components, and the proteolytic maturation of the protein was assessed by Western blotting using an antiserum against an EspP C-terminal peptide. No free β domain was detected, however, even after a prolonged incubation (**Figure 2—figure supplement 1D**).

Partly because we had difficulty reconstructing the Bam complex from BamAB and BamCDE sub-complexes, we next cloned the genes encoding all five subunits into a single expression plasmid. An octahistidine tag was introduced at the C-terminus of BamE to facilitate Bam complex purification. All of the proteins co-eluted on gel filtration columns (**Figure 2A,B**). This observation strongly suggests that Bam holocomplexes were formed efficiently in vivo. When the peak fractions were pooled and analyzed by Blue Native PAGE, a single band that migrated slightly slower than the 242 kDa marker was seen (**Figure 2C**). The Bam complex is only ~200 kDa, but it has previously been shown to migrate slower than its actual molecular weight on Blue Native gels (**Hagan et al., 2013**). In the presence of *E. coli* phospolipids, almost all of the purified protein, which we designated BamABCDE, could be reconstituted into proteoliposomes (**Figure 2D**). Remarkably, the intensity of the fluorescent signal in the fluorogenic peptide cleavage assay indicated that OmpT folded much more efficiently in the

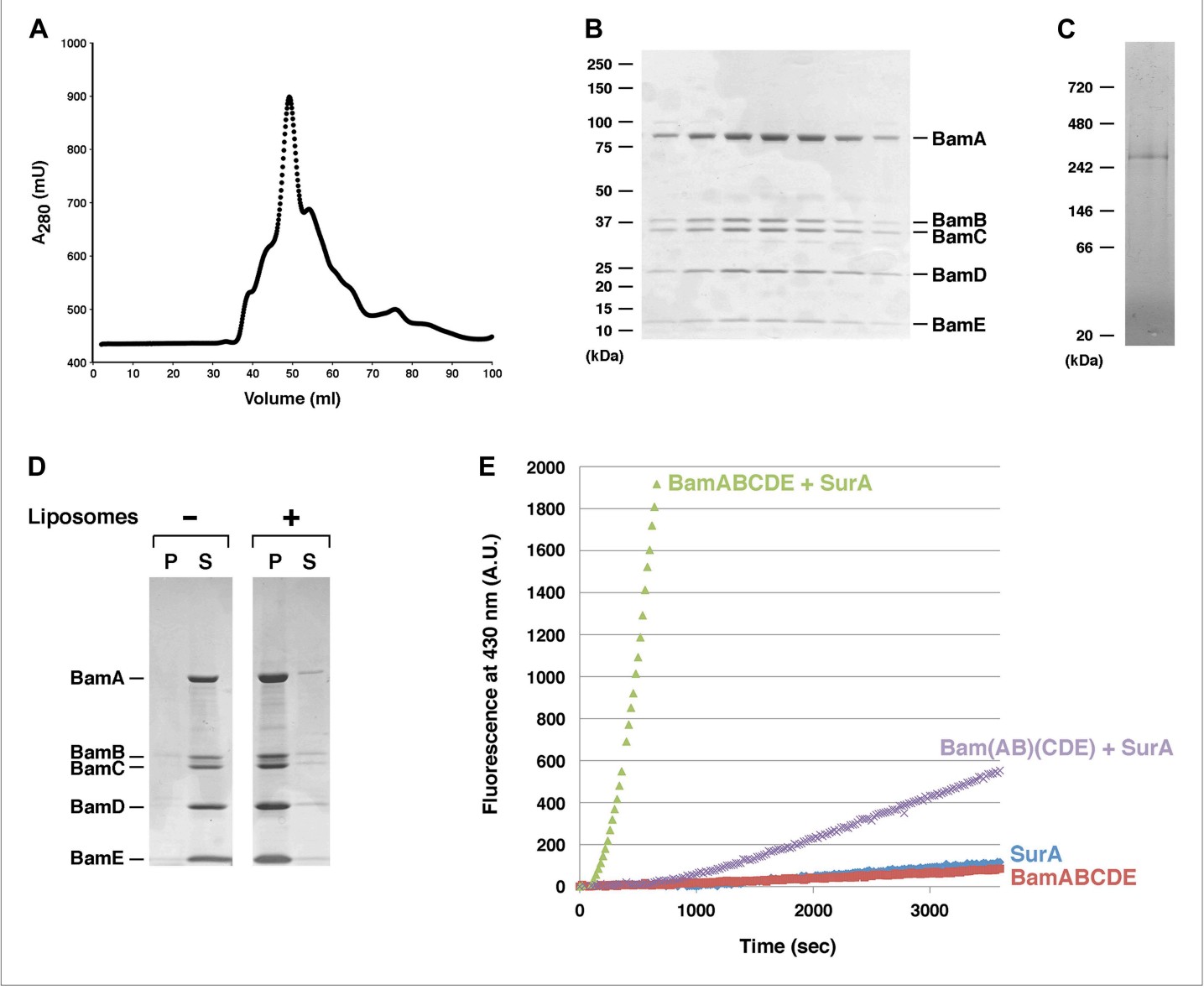

**Figure 2**. Purification and functional test of BamABCDE. (**A**) Chromatogram of BamABCDE on S-200 gel filtration column. (**B**) SDS-PAGE analysis of the peak fractions in (**A**). Proteins were visualized by Coomassie Blue staining. (**C**) The peak fractions in (**B**) were pooled and analyzed by Blue Native PAGE. (**D**) Purified BamABCDE was centrifuged in the absence of liposomes or after reconstitution into liposomes. The pellet (P) and supernatant (S) fractions were analyzed by SDS-PAGE. (**E**) Urea denatured OmpT was diluted and incubated with SurA and proteoliposomes containing either BamABCDE (green) or Bam(AB)(CDE) (purple), proteoliposomes containing BamABCDE alone (red) or SurA alone (blue). OmpT activity was assessed by measuring the fluorescent signal generated by the cleavage of a fluorogenic peptide.

The following figure supplements are available for figure 2:

**Figure supplement 1**. Test of purified Bam(AB)(CDE) in EspP assembly assay.

**Figure supplement 2**. SDS-PAGE analysis of purified SurA.

presence of BamABCDE than Bam(AB)(CDE) (*Figure 2E*, compare green and purple curves). These results suggested that the activity of the Bam complex is optimized when all of the subunits are expressed together.

Interestingly, we found that BamABCDE also stimulated the assembly of EspP(46+β). The urea-denatured EspP derivative was incubated at 30°C with proteoliposomes containing the Bam complex

and SurA, and samples were removed at various time points. Folding of the protein and subsequent proteolytic maturation were monitored by Western blot using the anti-EspP C-terminal antiserum and a fluorescently-labeled secondary antibody. A rapidly migrating (~27 kD) band that corresponds to the folded form of the free EspP β domain appeared within 5 min and was more prominent at later time points when the samples were not heated (*Figure 3A*). When the samples were heated, a much more intense ~30 kD band that corresponds to the unfolded β domain was observed. This marked difference in intensity has been observed previously (*Barnard et al., 2007*; *Pavlova et al., 2013*) and presumably results from the occlusion of the C-terminal epitope in the folded form of the β domain. No proteolytic processing was observed if either SurA or BamABCDE was omitted from the reaction or if BamAB or BamCDE subomplexes were used in place of the holocomplex (*Figure 3B*). Single plasmids were also used to produce Bam complexes that lack either BamB or BamC, but curiously only ~50–70% of the purified partial complexes could be reconstituted into proteoliposomes (*Figure 3—figure supplement 2A*). BamABDE and BamACDE both promoted the assembly EspP(46+β), but less effectively than BamABCDE (*Figure 3—figure supplement 2B*). Finally, the addition of purified Skp to reactions containing BamABCDE and SurA inhibited EspP(46+β) assembly (*Figure 3—figure supplement 3*) and ATP (0.1 mM) blocked assembly altogether (data not shown).

To confirm that the ~30 kDa fragment we observed was identical to the polypeptide that results from proteolytic maturation in vivo, we examined the assembly of a non-cleavable version of EspP(46+β). This derivative, which is designated EspP*(46+β), contains a mutation at the cleavage site that abolishes proteolysis but does not affect folding (*Skillman et al., 2005*). A polypeptide that migrated slower than the folded β domain (~30 kDa vs ~27 kDa) and that corresponds to the folded form of EspP*(46+β) was observed when the samples were not heated (*Figure 3C*). As expected, this band disappeared when the samples were heated because in the absence of cleavage the denatured form of EspP*(46+β) co-migrates with the unassembled protein.

Quantitation of the relative fluorescent signal produced by the denatured form of the EspP β domain and unprocessed EspP(46+β) on Western blots indicated that ~10–20% of the protein was typically assembled under our experimental conditions. The efficiency of assembly did not appear to be limited by the concentration of either reconstituted BamABCDE or SurA. A moderate increase in efficiency was observed when the BamABCDE concentration was increased from 0.05 μM to 0.2 μM (the concentration used in the experiments described here), but no increase was observed at higher concentrations (*Figure 3—figure supplement 4A*). Interestingly, if we added fresh substrate 5′ after the start of the incubation, the total yield of assembled EspP(46+β) increased in proportion to the amount of added substrate (*Figure 3—figure supplement 4B*). This observation is consistent with other evidence that the Bam complex can catalyze multiple rounds of assembly in vitro (*Hagan and Kahne, 2011*). In light of our results, it is likely that the efficiency of EspP(46+β) assembly was limited by the inherent ability of the denatured protein to remain assembly competent after dilution out of 8 M urea.

## The Bam complex and SurA catalyze the assembly of larger EspP derivatives

We next wished to determine whether the factors that promote the assembly of the EspP β domain can also promote the translocation of substantial passenger domain fragments. To this end, we expressed and purified EspP derivatives that contain progressively longer portions of the native passenger domain plus 20 N-terminal residues derived from the cloning vector (*Figure 3—figure supplement 1*). Because proteolytic maturation of EspP is absolutely dependent on the completion of passenger domain translocation in vivo (*Ieva and Bernstein, 2009*; *Peterson et al., 2010*; *Pavlova et al., 2013*), we initially assessed translocation by monitoring the appearance of the cleaved β domain. Indeed available evidence strongly suggests that the completion of β domain assembly (and the assembly of the active site) is a late event in autotransporter biogenesis (*Ieva et al., 2011*), possibly because the β domain forms at least part of the passenger domain transport channel (*Figure 1C*). As Western blot analysis using the anti-EspP C-terminal antiserum indicated, EspP derivatives that have passenger domains ranging in size from 72–734 residues were all processed with approximately the same efficiency as EspP(46+β) (*Figure 4*). The largest derivative lacks the N-terminal globular segment of the EspP passenger domain, but contains the entire β helical segment (*Figure 1A*). These results not only suggest that BamABCDE and SurA are sufficient to promote passenger domain translocation, but also suggest that the passenger domain does not readily adopt a translocation-incompetent conformation that independently limits the efficiency of autotransporter assembly.

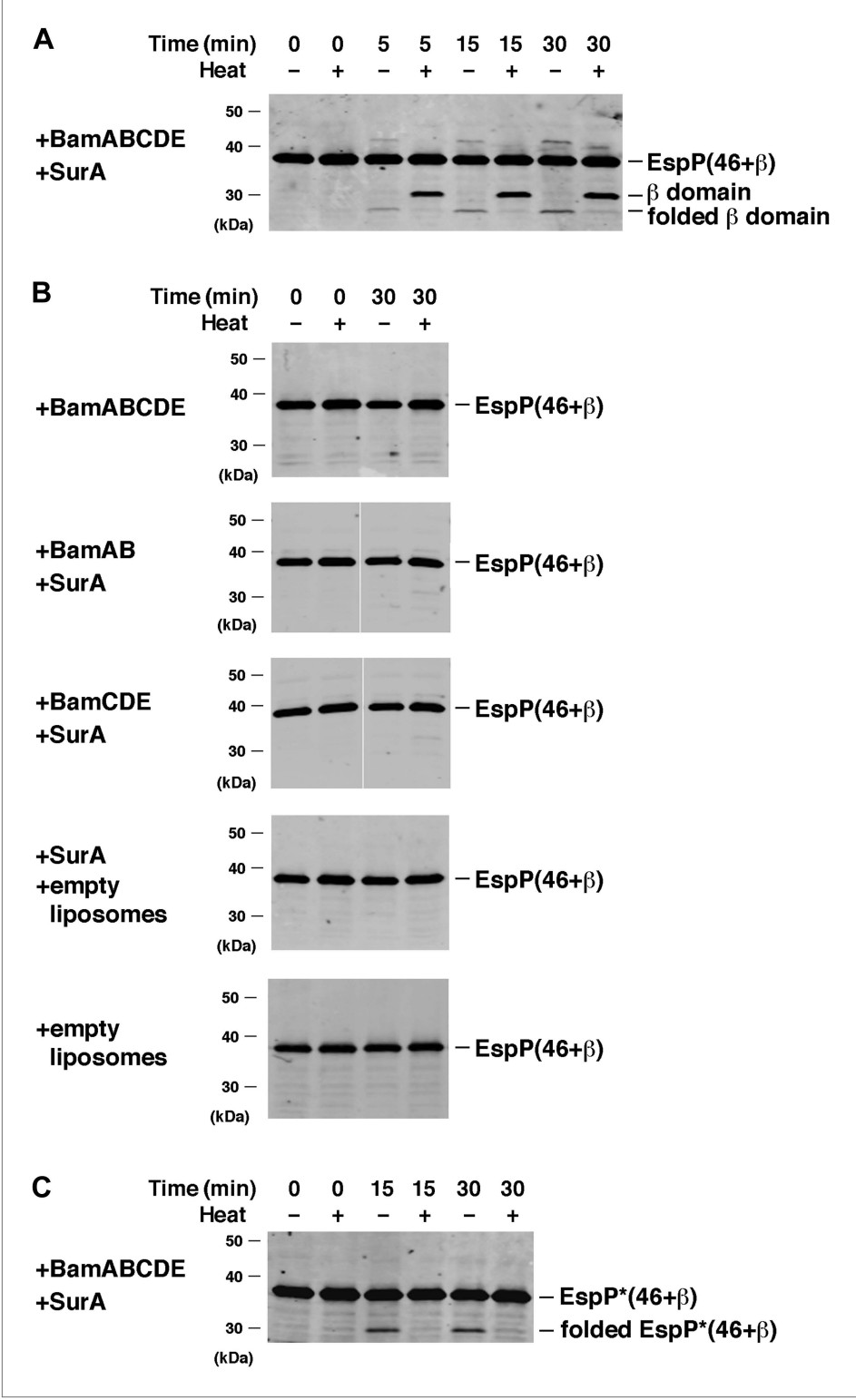

**Figure 3**. BamABCDE and SurA catalyze the assembly of EspP(46+β). (**A**) Urea-denatured EspP(46+β) was incubated with SurA and proteoliposomes containing BamABCDE. Aliquots were placed on ice at various time points, heated to 95°C or maintained at room temperature after the addition of SDS-PAGE buffer, and analyzed by Western blot using an anti-EspP C-terminal peptide antiserum. (**B**) Urea-denatured EspP(46+β) was incubated with the indicated factors. Aliquots were removed after 0 and 30 min and analyzed as in (**A**). (**C**) Urea-denatured

*Figure 3. Continued on next page*

*Figure 3. Continued*

EspP*(46+β) was diluted and incubated with SurA and proteoliposomes containing BamABCDE. Aliquots were removed at various time points and analyzed as in (**A**).

The following figure supplements are available for figure 3:

**Figure supplement 1**. SDS-PAGE analysis of purified EspP derivatives.

**Figure supplement 2**. Bam complexes lacking BamB or BamC catalyze the assembly of EspP(46+β) less effectively than the holocomplex.

**Figure supplement 3**. Skp inhibits the assembly of EspP(46+β).

**Figure supplement 4**. The efficiency of EspP(46+β) assembly is limited by the ability of the protein to remain folding-competent.

To obtain additional evidence that EspP derivatives that contain significant passenger domain fragments assemble correctly in the in vitro assay, we selected a few derivatives for further analysis. Initially we found that after incubating EspP(96+β) with proteoliposomes containing BamABCDE and SurA for 30 min, the cleaved β domain was both subject to heat denaturation in SDS and resistant to proteinase K (PK) digestion (*Figure 5A*). This result provided direct evidence that the β domain was not only properly folded, but also inserted into the proteoliposomes. Subsequently we examined the assembly of EspP*(96+β), a non-cleavable version of EspP(96+β), under the same conditions. In the absence of PK, a band that migrates slower than the folded β domain (~35 kDa vs ~27 kDa) and that corresponds to the folded form of the full-length protein was observed (*Figure 5B*). The finding that the ~35 kDa polypeptide was resistant to PK digestion unless detergent was added showed that the passenger domain was translocated into the lumen of the proteoliposomes. We also performed an initial analysis of the assembly of EspP(HA-251+β), a derivative that contains 251 residues of the EspP passenger domain plus an N-terminal HA tag (14 additional residues). After we incubated the protein with proteoliposomes containing BamABCDE and SurA we detected the β domain and the cleaved passenger domain on Western blots probed with anti-EspP C-terminal and anti-HA antisera, respectively, and both fragments were resistant to PK digestion unless detergent was added (*Figure 5C*). Because the epitope tag is the last segment of the protein that traverses the membrane, these results confirm that the translocation reaction went to completion.

## β domain assembly is the rate-limiting step in EspP biogenesis

To examine the biogenesis of EspP derivatives that have different length passenger domains in more detail, we next performed a kinetic analysis of the assembly of EspP(46+β) and EspP(HA-251+β). We used 0.2 µM substrate in these experiments because we found that doubling the EspP concentration moderately increased the efficiency of assembly. As suggested by a study on OmpT, the enhancement of assembly might be a fortuitous effect of increasing the concentration of urea, which presumably helps to maintain the folding-competence of the substrate (*Hagan and Kahne, 2011*). Western blot analysis using the anti-EspP C-terminal antiserum showed that EspP(46+β) underwent substantial proteolytic processing within 2 min (*Figure 6A*, top and *Figure 6B*, blue curve). Interestingly, EspP(HA-251+β) was processed at a very similar rate (*Figure 6C*, top and *Figure 6D*, blue curve). Indeed the rate constants obtained by fitting the assembly data for the two EspP derivatives to either a single exponential or two-step model were nearly identical (*Supplementary file 1*). In the case of the EspP(HA-251+β) derivative, a PK-resistant ~30 kD polypeptide that corresponds to the cleaved passenger domain accumulated in parallel with the cleaved β domain (*Figure 6E*, top). This observation confirmed that the passenger domain was translocated into the lumen of the proteoliposomes during the assembly reaction. We also introduced a mutation (G1066A) that slightly impairs folding of the β domain and thereby delays the initiation of passenger domain translocation by ~1 min in vivo (*Pavlova et al., 2013*) into EspP(46+β) and EspP(HA-251+β). The mutation caused a similar delay in the assembly of both EspP derivatives in vitro (*Figure 6* and *Supplementary file 1*). This observation corroborates the conclusion that defects in the folding of the β domain affect the ability of the Bam complex to catalyze subsequent steps in EspP assembly.

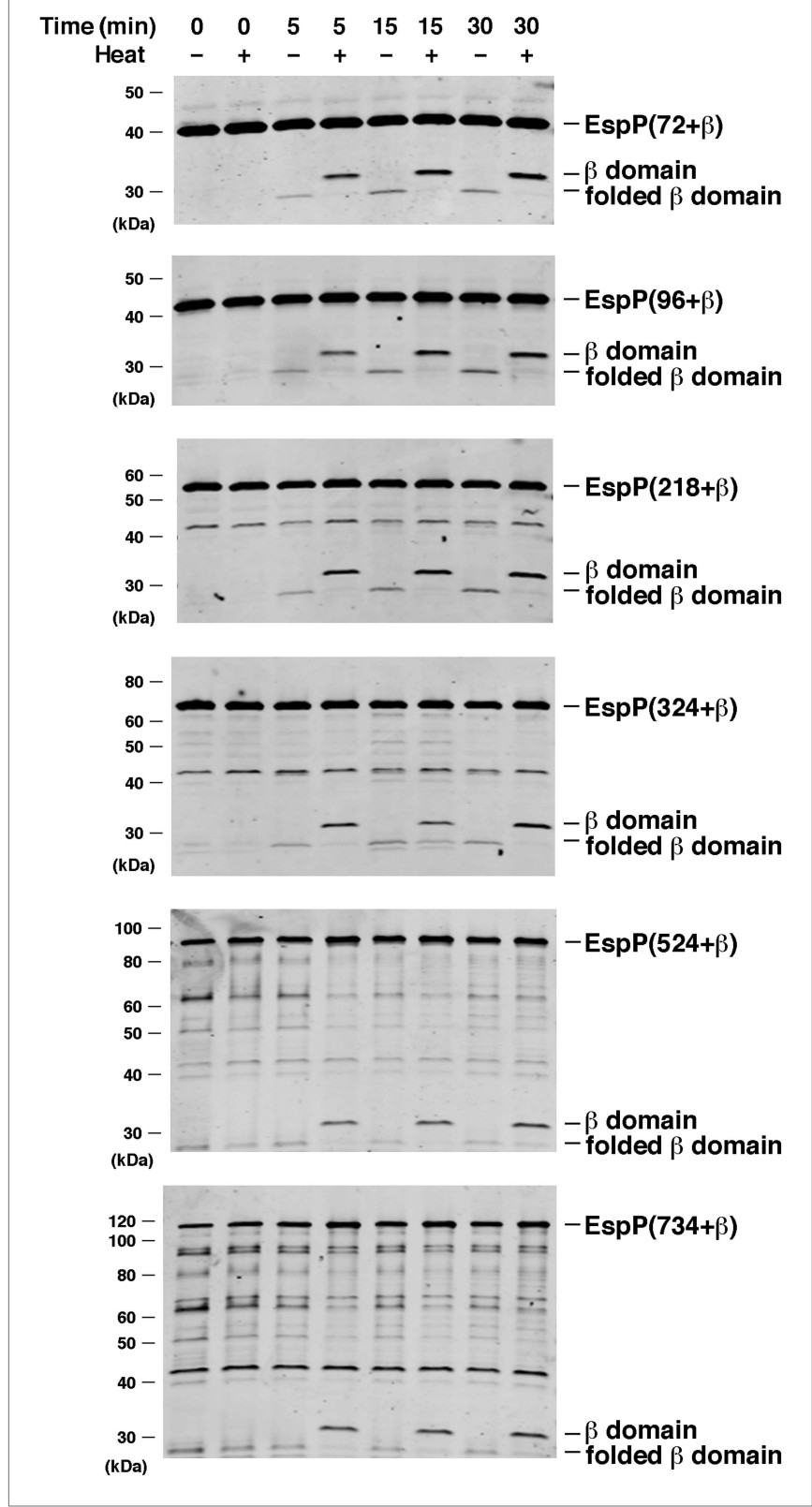

**Figure 4**. BamABCDE and SurA catalyze the assembly of longer EspP derivatives. The indicated urea-denatured EspP derivative was incubated with SurA and proteoliposomes containing BamABCDE. Aliquots were removed at various time points, heated to 95°C or maintained at room temperature after the addition of SDS-PAGE buffer, and

*Figure 4. Continued on next page*

*Figure 4. Continued*

analyzed by Western blot using an anti-EspP C-terminal peptide antiserum. The ~27 kDa polypeptide observed at the 0 min time point on the bottom two gels is an unidentified background band.

Interestingly, we found that an EspP derivative that contains an HA-tagged version of the full-length β helix [EspP(HA-714+β)] was also assembled very rapidly. Despite the presence of a much larger passenger domain, EspP(HA-714+β) was processed only slightly more slowly than EspP(46+β) (*Figure 7A,B*). Furthermore, the assembly data could be fit equally well to the same kinetic models and did not suggest the existence of an additional slow step (*Supplementary file 1*). A PK-resistant ~80 kDa polypeptide that corresponds to the cleaved passenger domain accumulated during the assembly reaction (*Figure 7C*). As expected, this polypeptide was degraded when the proteoliposomes were solubilized with detergent (*Figure 7D*). Taken together, the results indicate that passenger domain translocation is relatively fast, and that the assembly of the β domain is the rate-limiting step in autotransporter biogenesis. Curiously, the introduction of a short linker into EspP(HA-714+β) that impairs passenger domain folding and stalls translocation in vivo (*Ieva and Bernstein, 2009*) only modestly affected assembly in vitro (*Figure 7—figure supplement 1*). This observation, along with the finding that an EspP chimera containing a >200 residue intrinsically disordered passenger domain segment that is secreted efficiently in vivo could also assemble in the in vitro assay (*Figure 7—figure supplement 2*), suggests that in the absence of an exogenous energy source translocation is not driven exclusively by protein folding.

## Assembly of EspP into nanodiscs

Although it is often assumed that OM protein assembly is catalyzed by a Bam complex monomer, this notion has never been tested. Furthermore, given that autotransporter assembly is inherently more complicated than the assembly of generic β barrel proteins, it is conceivable that the Bam complex exists in a distinctive oligomeric state during its interaction with autotransporters. To determine whether a single copy of the Bam complex can mediate the assembly of EspP derivatives, we reconstituted BamABCDE into ~10–12 nm nanodiscs using conditions that facilitate the incorporation of a single copy of the ABC transporter MalFGK$_2$ (*Bao et al., 2012*). The nanodiscs eluted at about the same position on gel filtration columns as BamABCDE monomers (*Figure 8A*). Analysis of the peak fractions by SDS-PAGE suggested that the scaffold protein MSP1D1 was present in excess over the Bam complex proteins (*Figure 8B*). Because nanodiscs are stabilized by two copies of the scaffold protein (*Denisov et al., 2004*), however, this observation is consistent with the prediction that they contained a single copy of the Bam complex. In addition, the nanodiscs produced a single band on Blue Native PAGE that migrated only slightly slower than a BamABCDE monomer (compare *Figure 8C* to *Figure 1C*). Western blot analysis using the anti-EspP C-terminal antiserum revealed that both EspP(46+β) and EspP(HA-251+β) were assembled into nanodiscs about as efficiently as they were assembled into proteoliposomes (*Figure 8D*). Although the non-vesicular nature of nanodiscs precluded assessment of passenger domain translocation using a PK-sensitivity assay, we found that the ~30 kDa passenger domain of EspP(HA-251+β) accumulated in parallel with the cleaved β domain (*Figure 8E*). These results not only provide evidence that a Bam complex monomer is sufficient to catalyze autotransporter assembly, but also show that the curvature of the proteoliposome membrane is not itself required for the assembly process.

## Discussion

In this study we identified the minimal set of factors that are required for the complete assembly of the model autotransporter EspP. Initially we showed that the Bam complex reconstituted into proteoliposomes and SurA and both necessary and sufficient to catalyze the folding and proteolytic processing of a simplified version of EspP that consists of the β domain plus a short passenger domain fragment that protrudes only slightly from the β domain pore. This derivative is similar to generic β barrel proteins such as OmpT that reside almost entirely in the OM, except that it contains an internal α-helical segment. In subsequent experiments we found that the same two factors are sufficient to catalyze the translocation of passenger domain fragments associated with much larger EspP derivatives. Interestingly, the efficiency and kinetics of assembly was largely independent of the length of the derivative. While this observation does not exclude the possibility that there are other factors that

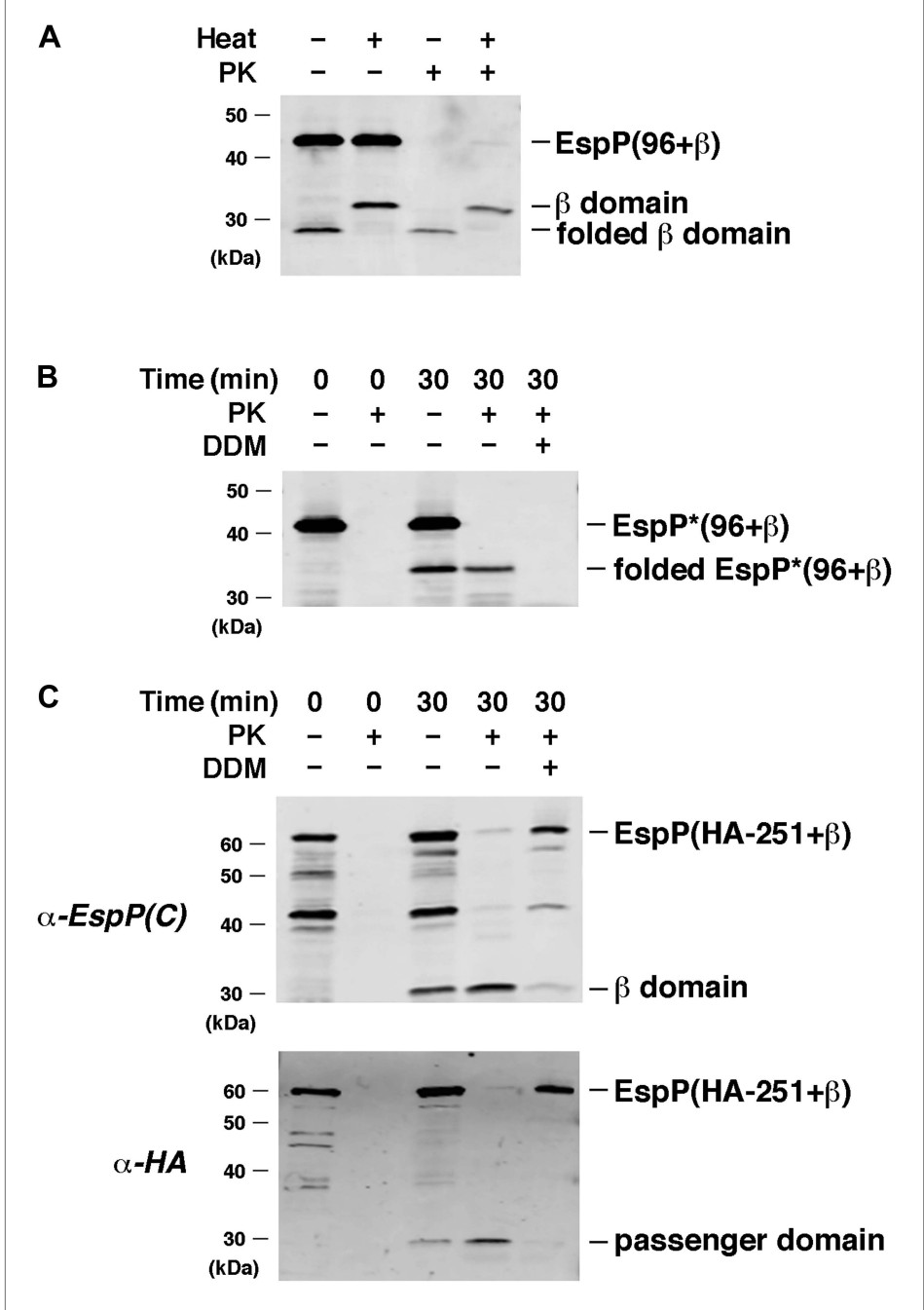

**Figure 5**. EspP derivatives are correctly assembled into proteoliposomes. (**A**) Urea-denatured EspP(96+β) was incubated with SurA and proteoliposomes containing BamABCDE. Aliquots were placed on ice after 0 and 30 min and either treated with PK or left untreated. After the addition of SDS-PAGE buffer samples were heated to 95°C or maintained at room temperature and analyzed by Western blot using an anti-EspP C-terminal peptide. (**B**) The experiment described in (**A**) was repeated with EspP*(96+β), except that n-dodecyl β-D-maltoside (DDM) was added to one sample prior to PK treatment, and none of the samples were heated after the addition of SDS-PAGE buffer. (**C**) The experiment described in (**A**) was repeated with EspP(HA-251+β), except that DDM was added to one sample prior to PK treatment, and all of the samples were heated to 95°C after the addition of SDS-PAGE buffer. Samples were divided in half and analyzed by Western blot using an anti-EspP C-terminal peptide antiserum or an anti-HA antiserum.

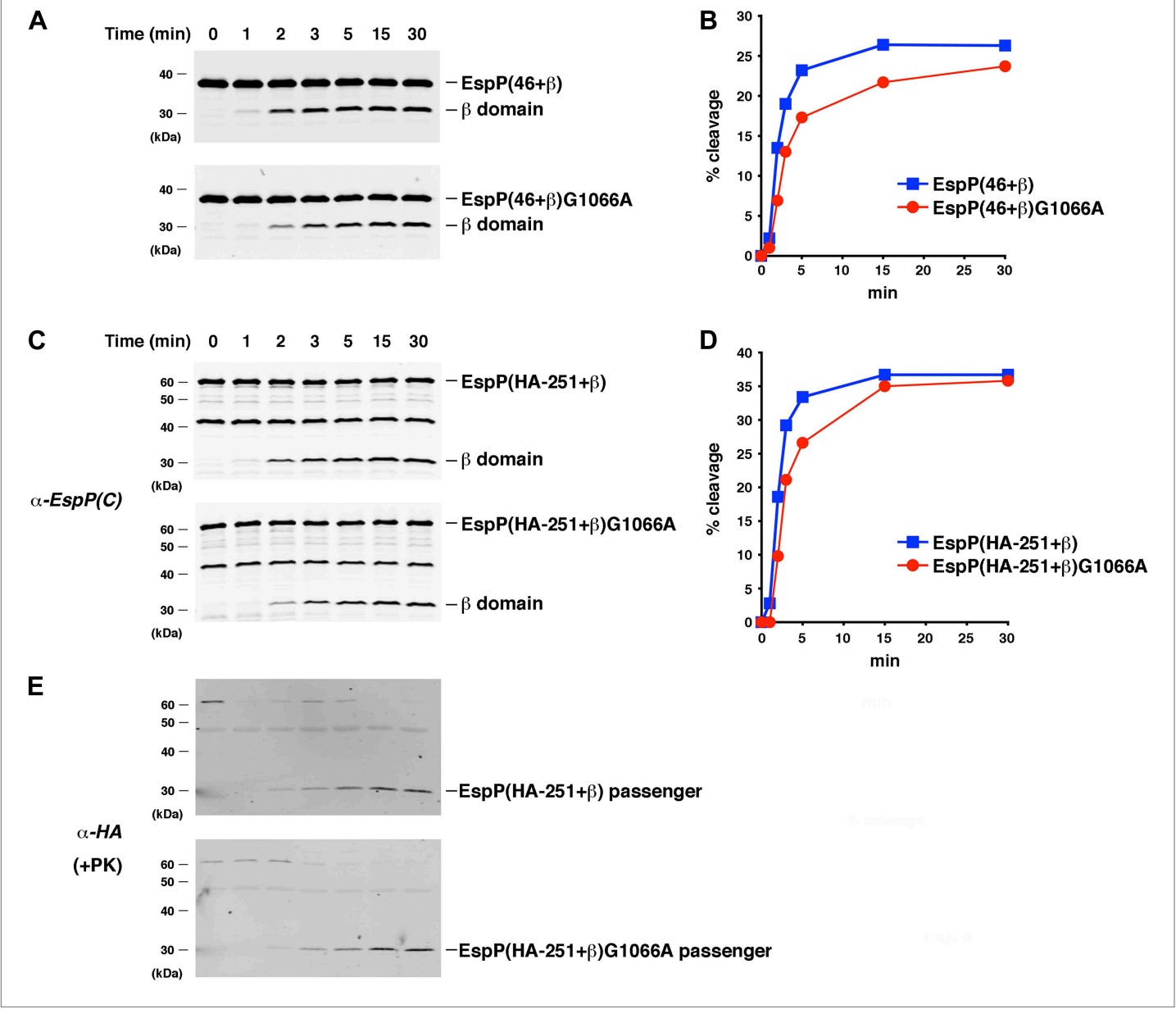

**Figure 6**. The assembly of the β domain is the rate-limiting step in EspP biogenesis. (**A**) Urea-denatured EspP(46+β) or EspP(46+β)G1066A was incubated with SurA and proteoliposomes containing BamABCDE. Aliquots were placed on ice at various time points, heated to 95°C after the addition of SDS-PAGE buffer, and analyzed by Western blot using an anti-EspP C-terminal peptide antiserum. (**B**) Quantitation of the data shown in (**A**). (**C**) The experiment shown in (**A**) was repeated with EspP(HA-251+β) or EspP(HA-251+β). (**D**) Quantitation of the data shown in (**C**). (**E**) A second aliquot from the experiment shown in (**C**) was placed on ice at each time point and treated with PK. After the addition of SDS-PAGE buffer the samples were heated to 95°C and analyzed by Western blot using an anti-HA antiserum.

stimulate passenger domain secretion in vivo, it does suggest that they would either play a limited role in the translocation reaction per se or serve to prevent misfolding in the crowded periplasmic space. Finally, experiments using nanodiscs provided evidence that EspP assembly is catalyzed by a single copy of the Bam complex. Overall, our results are consistent with a model in which the passenger domain is secreted through a channel consisting of the open β domain and/or the BamA β barrel in an open conformation (*Figure 1C*).

Our work demonstrates that passenger domain secretion does not require either an input of exogenous energy or the presence of IM proteins that transduce energy from the cytoplasm or the

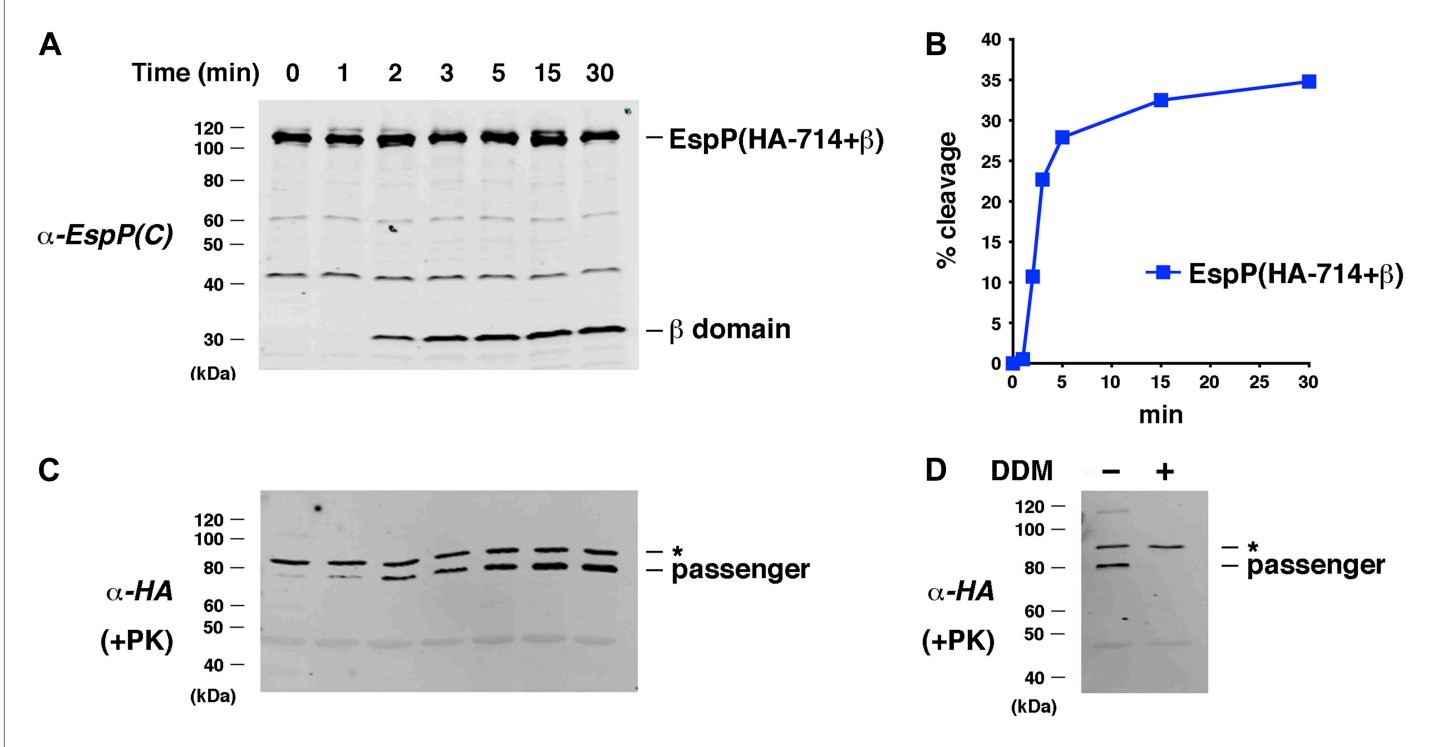

**Figure 7**. Assembly kinetics of an EspP derivative containing the full-length β helix. (**A**) Urea-denatured EspP(HA-714+β) was incubated with SurA and proteoliposomes containing BamABCDE. Aliquots were placed on ice at various time points, heated to 95°C after the addition of SDS-PAGE buffer, and analyzed by Western blot using an anti-EspP C-terminal peptide antiserum. (**B**) Quantitation of the data shown in (**A**). (**C**) A second aliquot from the experiment shown in (**A**) was placed on ice at each time point and treated with PK. After the addition of SDS-PAGE buffer the samples were heated to 95°C and analyzed by Western blot using an anti-HA antiserum. (**D**) The experiment described in (**A**) was repeated. Two equal aliquots were placed on ice after 30 min and DDM was added to one aliquot. The samples were then treated with PK and analyzed by Western blot as described in (**C**). In (**C** and **D**) the asterisk denotes an unidentified background band.

The following figure supplements are available for figure 7:

**Figure supplement 1**. Effect of a linker insertion in the passenger domain on the assembly of EspP(HA-714+β).

**Figure supplement 2**. BamABCDE and SurA catalyze the assembly of RD-EspP chimeras.

membrane potential. The results are surprising because ATP hydrolysis is required to drive protein translocation through the Sec complex in an analogous reconstituted assay system (*Brundage et al., 1990*) and because other types of protein translocation reactions appear to require a significant energy expenditure (*Alder and Theg, 2003*; *Shi and Theg, 2013*). The high energy cost may result in part from a tendency of proteins to slide backwards at specific stages of the translocation reaction (*Schiebel et al., 1991*; *Bauer et al., 2014*). In the autotransporter pathway, the folding of the passenger domain in the extracellular space has been shown to play a role in driving translocation in vivo, and folding may also promote translocation in the in vitro assay. Available evidence, however, neither supports the notion that stepwise folding alone drives translocation nor explains the efficient secretion of the intrinsically disordered RD domain or polypeptides that fold in the periplasm. As previously suggested (*Kang'ethe and Bernstein, 2013a*), electrostatic or other types of interactions between passenger domains and the Bam complex or membrane lipids may facilitate translocation. Like the FimD protein that secretes type I pilus subunits, the Bam complex might also guide passenger domains along a low-energy pathway (*Geibel et al., 2013*). Alternatively, the Bam complex might catalyze at least the initial stages of translocation and the assembly of the β domain in a concerted fashion. Indeed a model in which a polypeptide segment is pushed across the membrane during the membrane integration of the β domain might explain the efficient secretion of ~100 residue fragments that

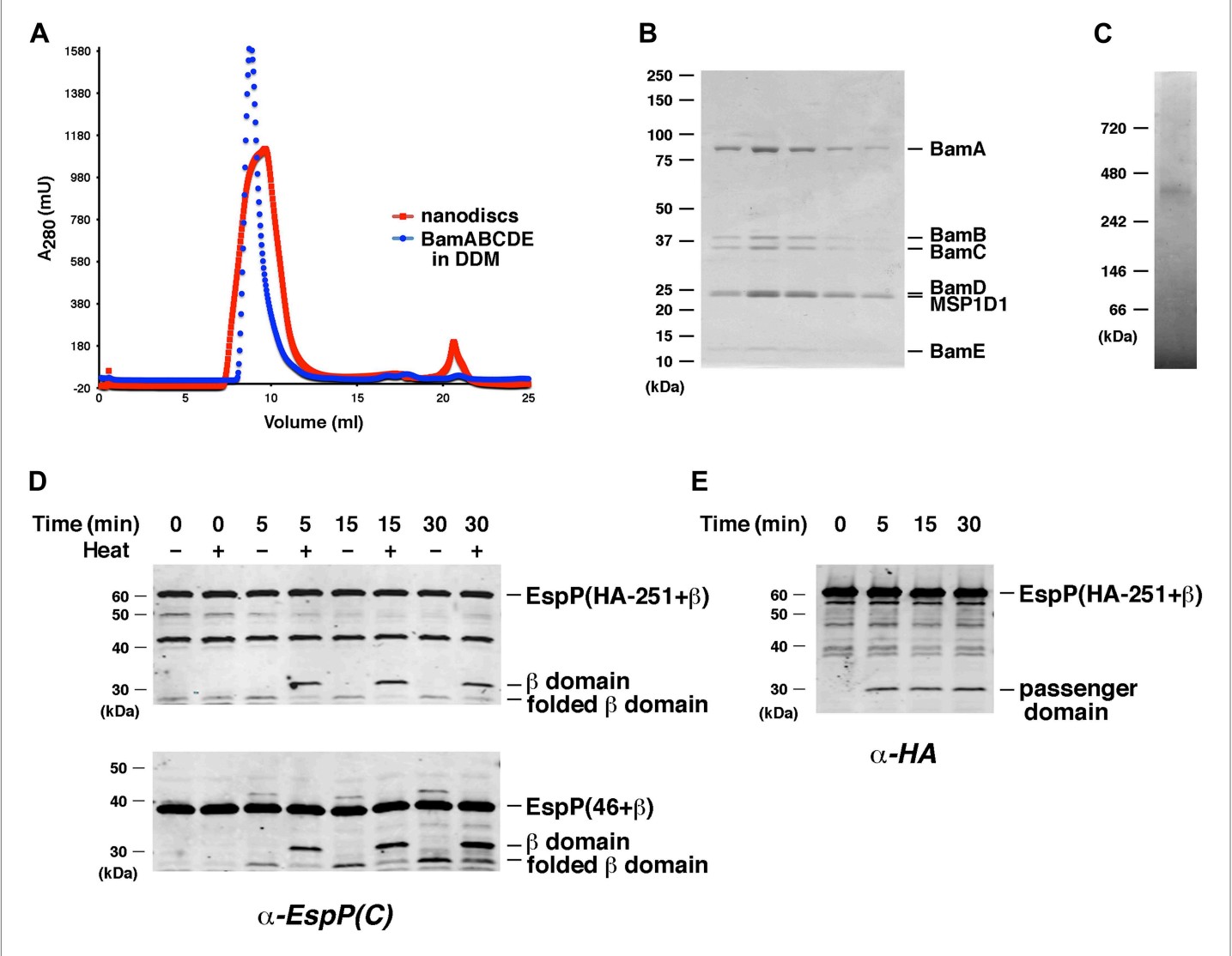

**Figure 8**. BamABCDE and SurA catalyze the assembly of EspP derivatives into nanodiscs. (**A**) Chromatograms of nanodiscs and BamABCDE in DDM on Superdex 75 gel filtration column. (**B**) SDS-PAGE analysis of the peak fractions in (**A**). Proteins were visualized by staining the gel with Coomassie Blue. (**C**) The peak fractions in (**B**) were pooled and analyzed by Blue Native PAGE. (**D**) Urea-denatured EspP(HA-251+β) or EspP(46+β) was incubated with SurA and nanodiscs containing BamABCDE. Aliquots were placed on ice at various time points, heated to 95°C or maintained at room temperature after the addition of SDS-PAGE buffer, and analyzed by Western blot using an anti-EspP C-terminal peptide antiserum. The ~27 kDa polypeptide observed at the 0 min time point on the top gel is an unidentified background band. (**E**) An additional aliquot from the EspP(HA-251+β) assembly reaction shown in (**D**) was removed at each time point, heated to 95°C after the addition of SDS-PAGE buffer, and analyzed by Western blot using an anti-HA antiserum.

are too short to fold (*Skillman et al., 2005*; *Pavlova, et al., 2013*). While the coupling of translocation to β domain assembly might also explain the assembly of a folding-deficient mutant such as EspP(HA-714+β)586TEV in vitro, the observation that stalled translocation reactions can be restarted in vivo (*Ieva and Bernstein, 2009*; JHP and HDB, unpublished results) strongly suggests that energy can be harnessed after the insertion of the β domain is largely complete.

Our results also indicate that the assembly of the β domain is the rate-limiting step in autotransporter biogenesis. Pulse-chase labeling and photocrosslinking experiments have previously provided evidence that β domain assembly is slower than passenger domain translocation in vivo (*Ieva et al., 2011*), but the interpretation of these experiments is complicated by the fact that the synthesis of the 1300 residue EspP protein is itself rather slow (~45 s). The use of fully synthesized EspP derivatives in the in vitro assay has enabled us to circumvent this problem and assess the contribution of the

translocation step to the overall reaction kinetics more effectively. It is noteworthy that the ability of the β domain to remain assembly-competent also appeared to limit the efficiency of the assembly reaction. This observation corroborates the conclusion that even long passenger domain segments fold slowly and consequently resist aggregation, at least in vitro (*Junker et al., 2006*). Furthermore, given that the β domain clearly interacts with Skp in vivo (*Ieva et al., 2011*; *Pavlova et al., 2013*), it is striking that the presence of the chaperone inhibited assembly in vitro. Presumably we could not recapitulate the productive interaction between Skp and the β domain that occurs in vivo, or a factor that is required to release Skp from client proteins was not present (or was present but not fully functional) in our assay.

Finally, the strategy that we have described to purify the Bam complex may facilitate future studies on the assembly of both autotransporters and many other OM proteins. It is striking that, at least in our hands, BamABCDE was more active than Bam(AB)(CDE). While this disparity may simply be due to technical issues, our results also raise the possibility that the structure of the Bam complex assembled in vivo differs from that of the Bam complex reconstructed from BamAB and BamCDE subcomplexes in vitro. Indeed it is conceivable that while a stable heterooligomer can be formed from the two sub-complexes, the Bam complex is actually assembled by a different pathway inside living cells. In this regard it should be noted that in preliminary experiments we obtained evidence that the Bam complex remains intact in vivo and does not undergo a dynamic cycle in which BamAB and BamCDE subcomplexes rapidly dissocate and reassociate (JHP and HDB, unpublished results).

## Materials and methods

### Plasmid construction

Previously described plasmids (*Hagan et al., 2010*) that encode the BamAB and BamCD subcomplexes, His-tagged BamE, SurA and the His-tagged OmpT G236K/K237G mutant (pSK38, pSK46, pBamE-His, pSK257 and pCH28) were reconstructed. To generate plasmid pJH113, a new Nde I site was introduced into the polylinker of a derivative of pTRC99a that lacks the endogenous Nde I site (*Szabady et al., 2005*) using the QuikChange Mutagenesis Kit (Agilent, Santa Clara, CA) with the oligonucleotide pTRC/Nde and its complement (all oligonucleotides used in this study are listed in *Supplementary file 2*). The genes that encode all five subunits of the Bam complex were then cloned sequentially into pJH113 to create plasmid pJH114. Each gene was amplified by PCR using genomic DNA from *E. coli* strain AD202 as a template (*Akiyama and Ito, 1990*). *BamA* was first cloned into the Nde I and BamH I sites of pJH113, and the other genes were then cloned into the BamH I site of the resulting plasmid. In the final round of cloning an octahistidine tag was added to the C terminus of BamE during PCR amplification. Plasmids pJH115, pJH116 and pJH117, which encode BamACDE(His$_8$), BamABDE(His$_8$), and BamCDE(His$_8$), respectively, were constructed in the same fashion except that one or more of the cloning steps were omitted. Plasmid pJH118, which encodes (His$_6$)BamAB, was made by first introducing an Eag I site into pJH113-*bamA* using oligonucleotide BamA.Eag(+) and its complement. A hexahistidine tag was then introduced at the N-terminus of the mature region of BamA using oligonucleotides BamAHis(+) and BamAHis(−). Finally, a Kpn I-Xba I fragment encoding the C-terminus of BamA and BamB was excised from pJH113-*bamAB* and cloned into the cognate sites of this plasmid. A plasmid encoding the MSP1D1 protein was previously described (*Denisov et al., 2004*) and was obtained from Addgene (plasmid 20061).

To make plasmids that express His$_6$-tagged EspP derivatives, fragments of *espP* were amplified by PCR using the primer EspP(−) and an appropriate upstream oligonucleotide and pRLS5, pJH62 or pKMS3 as a template (*Skillman et al., 2005*; *Szabady et al., 2005*). The resulting PCR products were digested with Nde I and BamH I and cloned into the cognate sites of pET28b. To construct plasmids encoding HA-tagged EspP derivatives, the oligonucleotides HA tag(+) and HA tag(−) (*Supplementary file 2*) were cloned into the Nco I and Nde I sites of plasmids encoding EspP(271+β) and EspP(734+β). To make plasmids expressing RD-EspP chimeras, fragments of the RD gene were amplified using appropriate primers and pWK2 (*Kang'ethe and Bernstein, 2013a*) as a template. PCR products were then digested with Nco I and Nde I and cloned into the cognate sites of pET28b encoding EspP(51+β).

### Assembly of the Bam complex from BamAB and BamCDE subcomplexes

BamAB and BamCDE(His$_8$) were produced independently essentially as described (*Hagan et al., 2010*). BL21(DE3) transformed with pSK38 were grown at 37°C in a 1 l volume to OD$_{600}$ = 0.3. Cultures were shifted to 25°C over 30 min and *bamAB* overexpression was induced by the addition of 0.1 mM

IPTG at $OD_{600}$ = 0.5–0.6. Cells were then incubated in the presence of the inducer for 3 hr. BL21(DE3) transformed with both pSK46 and pBamE-His were grown at 37°C in a 1 l volume to $OD_{600}$ = 0.5–0.6. *BamCD* and *bamE(His₈)* expression was induced by the addition of 0.1 mM IPTG, and incubation was continued for an additional 3–4 hr. Cells were pelleted in a Beckman JLA-8.1000 rotor (4000×*g*, 15 min), resuspended in 10 ml cold 20 mM Tris–HCl pH 8 and lysed using an Avestin EmulsiFlex C3 (3–4 passes). Lysates were centrifuged at 5000×*g* at 4°C for 10 min to remove unbroken cells. The supernatants were centrifuged in a Beckman Ti70 rotor (50,000 rpm, 30 min, 4°C) to pellet the membranes, which were solubilized in 10 ml TBS (25 mM Tris, pH 7.4137 mM NaCl, 3 mM KCl) containing 2% Triton X-100, 10 mM EDTA, and 10 µg/ml lysozyme for 30 min at room temperature. The ultracentrifugation step was repeated and the resulting supernatants were dialyzed overnight against cold buffer A (TBS/0.5% Triton X-100). BamAB was then further enriched by gel filtration on an S-200 column (GE Healthcare) equilibrated with buffer A. Partially purified BamAB and BamCDE were mixed in at least a 5:1 ratio and rotated in the presence of 2 ml Ni-NTA agarose for 1 hr at 4°C. The Ni-NTA beads were washed with one column volume buffer A containing 50 mM imidazole. The assembled Bam(AB)(CDE) complex was eluted with 3.5 ml buffer A containing 500 mM Imidazole and injected into an S-200 column equilibrated with TBS pH 8, 0.03% n-dodecyl-β-D-maltoside (DDM), 1 mM tris(2-carboxyethyl)phosphine (TCEP). The column was run at 0.5 ml/min and 1 ml fractions were collected. Portions of each fraction were analyzed by SDS-PAGE to identify fractions that contained complete Bam complexes. Typically 5–7 fractions were pooled and concentrated ~10-fold using Amicon Ultra-15 centrifugal filters (Millipore, Billerica, MA). The concentration of the purified Bam complex was then determined using the Bio-Rad *DC* Protein Assay following the manufacturer's instructions.

### Expression and purification of SurA

His-tagged SurA was overproduced and purified essentially as described (*Hagan et al., 2010*). In brief, 1 l cultures of BL21(DE3) transformed with pSK257 were grown at 37°C to $OD_{600}$ = 1 and shifted to 16°C. Cultures were incubated overnight after the addition of 0.1 mM IPTG (0.1 mM). Cells were then harvested as described above, resuspended in 10 ml cold 20 mM Tris HCl pH 8, and lysed using an EmulsiFlex C3. Lysates were clarified by centrifugation at 35,000×*g* at 4°C for 20 min. SurA was purified from the resulting supernatant using two consecutive rounds of TALON affinity chromatography (Clontech, Mountain View, CA) following the manufacturer's instructions. The purified protein was dialyzed overnight against 20 mM Tris–HCl pH 8 to remove the imidazole.

### Expression and purification of BamABCDE

*E. coli* strain HDB150 (MC4100 *ompT::spc ΔaraBAD leuD::kan*) transformed with pJH114 was grown overnight at 37°C in LB containing 100 µg/ml ampicillin. The cells were washed and diluted 1:80 into 1–2 l of fresh medium. When the cultures reached $OD_{600}$ = 0.5–0.6, 0.4 mM IPTG was added to induce the expression of *bamABCDE*. Cells were grown in the presence of the inducer for 1.5 hr and then harvested as described above. Cell pellets were resuspended in 10 ml/l cold 20 mM Tris–HCl pH 8 and cells were lysed using an EmulsiFlex C3. Lysates were centrifuged at 6000×*g* at 4°C for 10 min. The supernatants were centrifuged in a Ti 70 rotor as described above to isolate total membranes. After the pellets were incubated in 10 ml/l cold 50 mM Tris pH 8, 150 mM NaCl, 1% DDM on ice for 1 hr the centrifugation step was repeated. Supernatants containing the soluble membrane proteins were then rotated in the presence of 2 ml/l Ni-NTA agarose for 1.5 hr at 4°C. Ni-NTA beads were washed with one column volume buffer B (50 mM Tris pH 8, 150 mM NaCl, 0.03% DDM) containing 50 mM imidazole. BamABCDE was then eluted in 3.5 ml buffer A containing 500 mM imidazole and injected onto a S-200 column equilibrated with buffer A. The column was run at 0.5 ml/min and 1 ml fractions were collected. Fractions that contained complete BamABCDE complexes were identified, pooled and concentrated, and the concentration of the purified protein was determined as described above. NativePAGE (Blue Native) 4–16% Bis-Tris gels (Life Technologies, Grand Island, NY) were run to verify that the complex was intact. The same protocol was used to produce and purify BamACDE, BamABDE and the BamCDE and BamAB subcomplexes used in *Figure 3B* and *Figure 3—figure supplement 2*, except that cells were transformed with pJH115, pJH116, pJH117 or pJH118.

### Reconstitution of the Bam complex into proteoliposomes

*E. coli* phospholipids (*E. coli* Polar Lipid Extract, Avanti Polar Lipids, Alabaster, AL) were suspended in water at a concentration of 20 mg/ml and sonicated until well dispersed. A portion of the phospholipid suspension (40 µl) was added to 200 µl of the purified Bam(AB)(CDE) or BamABCDE (20 µM) and incubated on ice

for 5 min. The mixture was then diluted with 4 ml of 20 mM Tris HCl pH 8 and incubated on ice for 30 min to reduce the detergent concentration and promote proteoliposome formation. The proteoliposomes were pelleted in a Beckman TLA100.4 rotor (50,000 rpm, 4°C, 30 min) and resuspended in 200 µl 20 mM Tris HCl pH 8. Aliquots of the proteoliposomes were flash frozen in liquid nitrogen and stored at −80°C.

## Expression and purification of MSP1D1

BL21(DE3) transformed with pMSP1D1 (*Denisov et al., 2004*) were grown in 1 l cultures at 37°C, and 0.5 mM IPTG was added at $OD_{600}$ = 0.7–0.8. Cultures were incubated for 3 hr after the addition of the inducer. Cells were harvested, resuspended in 10 ml cold 20 mM Tris–HCl pH 8 and lysed as described above, and lysates were centrifuged at 35,000×*g* at 4°C for 20 min. Clarified supernatants were mixed with 2 ml NiNTA agarose and rotated for 1.5 hr at 4°C. The beads were washed with one column volume buffer D (40 mM Tris HCl pH 8, 300 mM NaCl) containing 50 mM imidazole. MSP1D1 was then eluted with 5 ml buffer D containing 500 mM imidazole and dialyzed overnight against cold buffer E (20 mM Tris HCl pH 7.4, 100 mM NaCl, 0.5 mM EDTA). The purified protein was then flash frozen and stored at −80°C. The concentration of MSP1D1 was determined spectrophotometrically using a previously determined $\varepsilon_{280}$ value (http://sligarlab.life.uiuc.edu/nanodisc/protocols.html). Before the protein was used to assemble nanodiscs it was diluted to 0.3 mM in TSGD buffer (50 mM Tris HCl pH 8, 100 mM NaCl, 0.03% DDM, 10% glycerol) (*Bao et al., 2012*).

## Reconstitution of the Bam complex into nanodiscs

Initially 10–15 ml (dry volume) Bio-Beads (Bio-Beads SM-2 Adsorbent, Bio-Rad, Hercules, CA) were washed and stored at 4°C in TS buffer (50 mM Tris HCl pH 8, 50 mM NaCl) as previously described (*Bao et al., 2012*). Nanodisc reconstitution was performed in a final volume of 300 µl essentially as described (*Bao et al., 2012*) by adding 6 µM purified Bam complex, 18 µM MSP1D1 and 360 µM phospholipids (*E. coli* Polar Lipid Extract, Avanti Polar Lipids) to TSGD buffer. The mixture was then incubated with Bio-Beads (50 µl) overnight at 4°C on a rocking platform. The Bio-Beads were then allowed to settle out, and the supernatant was loaded onto a Superdex 75 gel filtration column equilibrated with buffer F (50 mM Tris HCl pH8, 100 mM NaCl, 10% glycerol). The column was run at a flow rate was 0.5 ml/min, and 0.5 ml fractions were collected. Peak fractions were pooled and concentrated to a volume of 300 µl using Amicon Ultra-15 centrifugal filters.

## Expression and purification of OM proteins

OmpT and all EspP derivatives were synthesized without their signal peptides and purified from inclusion bodies. BL21(DE3) transformed with an appropriate plasmid were grown overnight at 37°C in LB containing 50 µg/ml kanamycin and diluted 1:80 into 1 l fresh medium. When the cultures reached $OD_{600}$ = 0.7, 0.5 mM IPTG was added. The cultures were incubated in the presence of the inducer for 3 hr. The cells were then harvested, resuspended in 10 ml cold TBS, and lysed as described above. Lysates were centrifuged at 5000×*g* at 4°C for 10 min. Pellets containing the inclusion bodies were resuspended in 10 ml TBS and washed twice by repeating the centrifugation step. After the final wash, pellets were resuspended in 5 ml 8 M urea and incubated at room temperature for 1 hr. Samples were then chilled briefly on ice and centrifuged in a Beckman TLA100.4 rotor (26,000 rpm, 4°C, 20 min). The concentration of the overexpressed OM proteins, which were highly enriched in the supernatants, were determined using the Bio-Rad *DC* Protein Assay.

## OmpT folding assays

The folding of OmpT was monitored by slightly modifying a previously described assay (*Hagan et al., 2010*). OmpT was diluted from a 293 µM stock to a final concentration of 20 µM and incubated with SurA (final concentration 140 µM) in 50 µl 20 mM Tris pH 6.5 for 10 min at room temperature. The Bam complex and the fluorogenic peptide Abz-Ala-Arg-Arg-Ala-Tyr(NO$_2$)-NH$_2$ (New England Peptide, Gardner, MA) were diluted in 50 µl of the same buffer to a final concentration of 16 µM and 2 mM, respectively. The two sub-reactions were then mixed together and the increase in fluorescence at 30°C was monitored on a Spectramax M5 fluorescent plate reader for 1 hr with readings every 20 s. Fluorescence emission was recorded at 430 nm following excitation at 325 nm.

## EspP assembly assays

An appropriate EspP derivative, SurA and proteoliposomes containing Bam complex were added successively to 20 mM Tris HCl pH 8. Reaction components were mixed after each addition. In the

experiment shown in *Figure 3—figure supplement 3*, Skp (obtained as a highly purified native trimer from MyBioSource.com, San Diego, CA) was added at the indicated concentration before SurA. In general the final concentration of reaction components was 0.1 µM EspP (diluted from a 6 µM stock solution in 8 M urea), 1 µM SurA and 0.2 µM Bam complex. In the experiments shown in *Figures 6 and 7* and *Figure 7—figure supplement 1*, the EspP and SurA concentrations were raised to 0.2 µM and 2 µM, respectively. All assembly reactions were performed at 30°C. At each time point, aliquots were removed and either mixed with an equal volume of 2× SDS-PAGE sample buffer or incubated on ice for 5 min with PK (5 µg/ml). In some experiments, 1% DDM was added prior to PK treatment. Protease digestions were stopped by the addition of 1 mM PMSF and 2× SDS-PAGE sample buffer. To assess the sensitivity of the EspP β domain to SDS denaturation, half of each sample was heated at 95°C for 5 min while the other half was maintained at room temperature. Proteins were resolved by SDS-PAGE and transferred to nitrocellulose using an iBlot apparatus (Life Technologies). Western blotting was conducted using antisera generated against an EspP C-terminal peptide (*Szabady et al., 2005*) or the HA (Y-11) peptide (Santa Cruz Biotechnology, Dallas, TX). Antibody-antigen complexes were detected by incubating filters with an IRDye 680-conjugated goat anti-rabbit antiserum and monitoring fluorescence at 700 nM using an Odyssey infrared imaging system (Licor, Lincoln, NE). Percent substrate cleavage at each time point was defined as 100 × (cleaved β domain/unprocessed substrate + cleaved β domain).

### SDS-PAGE
Novex 8–16% minigels (Life Technologies) were used to monitor protein purifications and to resolve the products of all EspP assembly reactions.

### Kinetic analysis of EspP assembly
Data from EspP assembly experiments was fit to single exponential and lag-phase kinetic models using KaleidaGraph 4.1 (Synergy Software, Reading, PA). The percent substrate cleavage at the 30 min time point was defined as maximum assembly and was used to determine a rate constant $k_1$ (or two rate constants $k_1$ and $k_2$ for a sequential two-step model). As in classical enzyme kinetics, $t_{1/2} = \ln 2/k$.

## Acknowledgements

We would like to thank Shu-ou Shan for help fitting the data obtained in EspP assembly experiments to kinetic models and Franck Duong for advice on preparing nanodiscs. This work was supported by the Intramural Research Program of the National Institute of Diabetes and Digestive and Kidney Diseases.

## Additional information

### Funding

| Funder | Author |
| --- | --- |
| National Institute of Diabetes and Digestive and Kidney Diseases | Giselle Roman-Hernandez, Janine H Peterson, Harris D Bernstein |

The funder had no role in study design, data collection and interpretation, or the decision to submit the work for publication.

### Author contributions
GR-H, HDB, Conception and design, Acquisition of data, Analysis and interpretation of data, Drafting or revising the article, Contributed unpublished essential data or reagents; JHP, Acquisition of data, Contributed unpublished essential data or reagents

## Additional files

### Supplementary files
• Supplementary file 1. Kinetic analysis of EspP assembly in vitro.

• Supplementary file 2. Oligonucleotides used in this study.

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
