## [Decision Letter]

Thank you for sending your work entitled “Reconstitution of bacterial autotransporter assembly using purified components” for consideration at *eLife.* Your article has been favorably evaluated by Randy Schekman (Senior editor) and 3 reviewers, one of whom is a member of our Board of Reviewing Editors.

The Reviewing editor and the other reviewers discussed their comments before we reached this decision, and the Reviewing editor has assembled the following comments to help you prepare a revised submission.

The manuscript was felt to be an important advance that provides new insights into autotransporter assembly and opens up new directions for future study. The topic is important, and the results were judged to be of interest to the wide readership of *eLife.* Minor suggestions for improvement are provided in the unedited referee comments below. These can all be handled with changes to the manuscript text and do not appear to require additional experimental work. Based on these comments, I anticipate you should have no difficulty preparing a revised manuscript that should be acceptable for publication in *eLife*. We look forward to receiving the revision.

Reviewer #1:

This carefully performed and well written study describes the reconstitution of autotransporter assembly and function with purified factors composed of the Bam complex and SurA. The advance in this paper lies partially in its technical aspects and partially in the conceptual insights. The technical advance relates to the optimization of highly functional Bam complex purification and reconstitution, while the conceptual advances relate to defining the minimal factors for autotransporter assembly and resolving key issues such as lack of an energy requirement. There is little to criticize about the experiments shown in this study, and the technical advance should now make it possible to perform detailed mechanistic analysis, for example by applying biophysical assays for different steps and interactions in the pathway. The insights into autotransporter assembly appear to largely support contemporary models based on earlier in vivo and structural work.

Minor comments:

1) Is it worth including BamC and BamD in the diagram in Figure 1?

2) In the longest two constructs in Figure 4, the folded beta domain band is apparently confounded by a co-migrating background band, but only in the zero timepoint. This is worth pointing out so a reader isn't confused. Also, why does it disappear?

3) In some of the experiments with longer passengers, the starting protein preparation is a bit heterogeneous (e.g., Figure 4 or Figure 6). Do the authors know if these are N- or C-terminal truncations? How then do the authors know that the beta domain is arising only from the full length product?

4) In Figure 7, the passenger band in the HA blot seems to appear slightly before the beta domain. It may be worth plotting both on the same graph and explaining this apparent discrepancy with the model of passenger cleavage only occurring after beta domain assembly.

Reviewer #2:

Autotransporters are outer membrane proteins composed of a beta-barrel domain that anchors them to the outer membrane and a passenger domain that is translocated into the environment. This architecture led to models postulating that the beta-barrel domain was the translocator for the passenger domain and that the folding of the passenger domain drove translocation. The dependence of autotransporter biogenesis on the Bam (beta-barrel assembly machine) complex shifted the field of autotransporter biogenesis away from these models. Recently, the Bernstein lab has proposed a model stating that the beta-barrel domain is inserted in the outer membrane by the Bam complex and that the translocation of the passenger domain involves Bam and an intermediate in the folding of the beta-barrel domain. Furthermore, they have also proposed that the energy driving translocation is provided in part by the folding of the passenger domains and uncharacterized interactions involving acidic residues in the passenger domain. Much of this model is based on cross-linking experiments, which can be hard to interpret. In addition, it was not clear if other factors assist in this process. Here, using an in vitro reconstitution system, the Bernstein lab shows that the chaperone SurA and the Bam complex are required and sufficient for the assembly of the EspP autotransporter into proteoliposomes and nanodiscs. This is a major accomplishment and a significant contribution to the field that can be used to study the mechanistic details of autotransporter biogenesis. Indeed, the authors perform kinetic measurements and conclude that the assembly of the beta-barrel domain is the rate-limiting step in the biogenesis of this autotransporter. These findings are consistent with their aforementioned model of translocation of the passenger domain occurring during the assembly of the beta-domain.

Overall, the manuscript is well written, the data are of the very high quality, and the work constitutes solid step forward in autotransporter biogenesis research. Well done.

Minor comments:

The only minor issue I have with this work refers to the fact that the authors highlight the development of a novel method to purify a highly active form of the Bam complex (e.g., in the Abstract). Their new method is based on the co-expression and co-purification of all 5 components of the Bam complex, as opposed to the separate expression of two sub-complexes of two and three Bam members that are later assembled together. It is clear that in their hands their new method yields a more active Bam complex. I assume this is a reproducible finding across different preparations, but I am not sure if this is a lab-specific phenomenon or not. I agree with the two possible explanations (technical issue vs biological significant) given in the last paragraph of the Discussion, although I recommend the authors remove the last sentence as it is based on being unable to detect something. Their comparison of the two methods should remain in the manuscript but I suggest that the authors tone down the development of a novel method (for example, remove the last sentence of the Abstract).

Reviewer #3:

This manuscript by Roman-Hernandez, Peterson & Bernstein describes experiments that show that the Bam complex plus SurA are sufficient for assembly of a model autotransporter protein (EspP) into (outer) membranes. The authors describe a method for producing a purified Bam complex that has increased function over Bam complexes produced using previous purification/reconstitution methods. They use this Bam complex with purified SurA to demonstrate that these proteins are sufficient for assembly of EspP proteins into proteoliposomes and PK-digestion experiments provide evidence that the passenger domains are translocated across the membranes. The results demonstrate that no external energy is needed for the assembly/translocation process. The results are supported by experiments using nanodiscs and the nanodisc experiments provide compelling evidence that a single copy of the Bam complex is sufficient to translocate EspP. The manuscript is extremely well written and easy to follow, which is especially commendable given the complexity of the experiments and the data. The experimental design is appropriate for the questions being addressed, appropriate controls were included, and conclusions drawn were justified by the data. The results represent a large step forward in understanding how proteins of the autotransporter family are inserted into the outer membrane in their proper orientation and in membrane protein biogenesis in general. The results, and the approaches used in this manuscript, will be of interest to a broad audience.

Specific criticism:

1) The Results described in the section “…the introduction of a short linker into EspP…” are somewhat unsatisfying. Up to this point in the manuscript, the data suggest that the in vitro system reflects what happens in vivo. However, here, two proteins that show altered or impaired translocation in vivo are able to be assembled/translocated in vitro almost as efficiently as 'wild-type' proteins. Do these results mean that the in vitro system does not accurately reflect what is happening in vivo? The authors touch on this point only briefly in their discussion – I think this point should be expanded a bit.

Minor criticisms:

1) In the Introduction, the authors discuss autotransporter proteins as if they were all virulence factors. Although most that have been studied do play roles in pathogenesis, analyses have been skewed by the fact that most bacteria in which AT proteins have been studied are pathogens. It is highly likely that many, if not most, AT proteins perform other functions (and hence the authors' work is more broadly relevant than suggested by the Introduction).

2) The authors should add a reference to the new paper by Noinaj et al in Structure. (Also note that Buchanan was misspelled in the [28] reference.)

---

## [Author Response]

Reviewer #1:

Minor comments:

*1) Is it worth including BamC and BamD in the diagram in*
Figure 1*?*

We have omitted BamC and BamE to prevent the diagram from become cluttered and potentially confusing. To address the reviewer’s concern we now note in the Figure legend that BamC and BamE have been omitted for the sake of clarity.

*2) In the longest two constructs in*
Figure 4*, the folded beta domain band is apparently confounded by a co-migrating background band, but only in the zero timepoint. This is worth pointing out so a reader isn't confused. Also, why does it disappear?*

As suggested, we now state in the legends to Figure 4 and Figure 8 that the ∼27 kDa band that appears at the 0 min time point is a background band. We really do not know why it disappears. One possibility is that the protein is a C-terminal truncation of EspP that forms high molecular weight aggregates during the course of the reaction.

*3) In some of the experiments with longer passengers, the starting protein preparation is a bit heterogeneous (e.g.,*
Figure 4
*or*
Figure 6*). Do the authors know if these are N- or C-terminal truncations? How then do the authors know that the beta domain is arising only from the full length product?*

These polypeptides are presumably C-terminal truncations because they are detected with an antiserum raised against a C-terminal EspP peptide, but they might also be cross-reactive contaminants. We cannot rule out the possibility that some of the truncated protein was assembled during the reaction. Because there was no correlation between the presence of truncated polypeptides and the efficiency of assembly, however, most of the cleaved β domain was likely generated from the full-length protein.

*4) In*
Figure 7*, the passenger band in the HA blot seems to appear slightly before the beta domain. It may be worth plotting both on the same graph and explaining this apparent discrepancy with the model of passenger cleavage only occurring after beta domain assembly*.

The reviewer is correct, but the slightly earlier appearance of the passenger domain is a minor quirk of the experiment that is shown and not a reproducible result that would be worth emphasizing.

Reviewer #2:

Minor comments:

*The only minor issue I have with this work refers to the fact that the authors highlight the development of a novel method to purify a highly active form of the Bam complex (e.g., in the Abstract). Their new method is based on the co-expression and co-purification of all 5 components of the Bam complex, as opposed to the separate expression of two sub-complexes of two and three Bam members that are later assembled together. It is clear that in their hands their new method yields a more active Bam complex. I assume this is a reproducible finding across different preparations, but I am not sure if this is a lab-specific phenomenon or not. I agree with the two possible explanations (technical issue vs biological significant) given in the last paragraph of the Discussion, although I recommend the authors remove the last sentence as it is based on being unable to detect something. Their comparison of the two methods should remain in the manuscript but I suggest that the authors tone down the development of a novel method (for example, remove the last sentence of the Abstract)*.

To address the reviewer’s concern we have deleted the word “novel” from the Abstract and the last paragraph of the Discussion. We believe that this change enables us to draw attention to the fact that we used a method to purify the Bam complex that is different from the one that was previously described without suggesting that our method is better or completely novel. We have also substantially modified the last sentence of the Discussion to emphasize the positive aspect of our preliminary results.

Reviewer #3:

*[…] The results demonstrate that no external energy is needed for the assembly/translocation process. The results are supported by experiments using nanodiscs and the nanodisc experiments provide compelling evidence that a single copy of the Bam complex is sufficient to translocate EspP. The manuscript is extremely well written and easy to follow, which is especially commendable given the complexity of the experiments and the data. The experimental design is appropriate for the questions being addressed, appropriate controls were included, and conclusions drawn were justified by the data. The results represent a large step forward in understanding how proteins of the autotransporter family are inserted into the outer membrane in their proper orientation and in membrane protein biogenesis in general. The results, and the approaches used in this manuscript, will be of interest to a broad audience*.

Specific criticism:

*1) The Results described in the section “…the introduction of a short linker into EspP…” are somewhat unsatisfying. Up to this point in the manuscript, the data suggest that the in vitro system reflects what happens in vivo. However, here, two proteins that show altered or impaired translocation in vivo are able to be assembled/translocated in vitro almost as efficiently as 'wild-type' proteins. Do these results mean that the in vitro system does not accurately reflect what is happening in vivo? The authors touch on this point only briefly in their Discussion – I think this point should be expanded a bit*.

In light of the reviewer’s comments we now realize that the passage was slightly confusing. We found that only one EspP derivative that was assembled in the in vitro assay shows impaired translocation in vivo [EspP(HA-714+β)TEV586]. The second protein that the reviewer alludes to (the RD-EspP chimera) is assembled efficiently in vivo. We have now revised the text to clarify this point. Although we discuss the behavior of EspP(HA-714+β)TEV586 in the in vitro assay, we really do not know why the linker insertion exerts a stronger effect on the assembly of this EspP derivative in vivo. Of course our assay, like any in vitro assay, may not perfectly replicate all aspects of a biological process. As we note, our results suggest that a factor (or perhaps an activity of SurA or the Bam complex) that promotes the dissociation of EspP from the molecular chaperone Skp in vivo may be missing from the current assay system.

Minor criticisms:

*1) In the Introduction, the authors discuss autotransporter proteins as if they were all virulence factors. Although most that have been studied do play roles in pathogenesis, analyses have been skewed by the fact that most bacteria in which AT proteins have been studied are pathogens. It is highly likely that many, if not most, AT proteins perform other functions (and hence the authors' work is more broadly relevant than suggested by the Introduction)*.

The reviewer raises an excellent point here, The reviewer raises an excellent point here. To avoid getting bogged down in a discussion of the function of autotransporters (which is not the focus of our study) we now simply state that the passenger domain “often mediates a virulence function”. We hope that readers will understand this to mean that autotransporters can also perform other functions.

*2) The authors should add a reference to the new paper by Noinaj et al in Structure. (Also note that Buchanan was misspelled in the*
[28]
*reference*.*)*

We have added the suggested reference and corrected the spelling mistake.